# Prevalence of Chronic Pulmonary Aspergillosis in Patients Suspected of Chest Malignancy

**DOI:** 10.3390/jof8030297

**Published:** 2022-03-13

**Authors:** Rasmus Rønberg, Jesper Rømhild Davidsen, Helmut J. F. Salzer, Eva Van Braeckel, Flemming Schønning Rosenvinge, Christian B. Laursen

**Affiliations:** 1Odense Respiratory Research Unit (ODIN), Department of Clinical Research, University of Southern Denmark, 5000 Odense, Denmark; jesper.roemhild.davidsen@rsyd.dk (J.R.D.); christian.b.laursen@rsyd.dk (C.B.L.); 2Odense Patient Data Explorative Network, Odense University Hospital, 5000 Odense, Denmark; 3Department of Respiratory Medicine, Odense University Hospital, 5000 Odense, Denmark; 4South Danish Center for Interstitial Lung Diseases (SCILS), Odense University Hospital, 5000 Odense, Denmark; 5Pulmonary Aspergillosis Center Denmark (PACD), Odense University Hospital, 5000 Odense, Denmark; 6Department of Internal Medicine/Pulmonary Medicine, Kepler University Hospital, 4020 Linz, Austria; salzer.helmut@gmail.com; 7Department of Respiratory Medicine, Ghent University Hospital, 9000 Ghent, Belgium; eva.vanbraeckel@ugent.be; 8Department of Internal Medicine and Paediatrics, Ghent University, 9000 Ghent, Belgium; 9Department of Clinical Microbiology, Odense University Hospital, 5000 Odense, Denmark; flemming.rosenvinge@rsyd.dk; 10Research Unit of Clinical Microbiology, Department of Clinical Research, University of Southern Denmark, 5000 Odense, Denmark

**Keywords:** chronic pulmonary aspergillosis, lung cancer, *Aspergillus*, prevalence

## Abstract

Chronic pulmonary aspergillosis (CPA) is a potentially life-threatening fungal lung infection, and recent research suggests CPA to be more common than previously considered. Although CPA mimics other lung diseases including pulmonary cancer, awareness of this disease entity is still sparse. This study aimed to investigate the prevalence of CPA in a population of patients under suspicion of having lung cancer. We conducted a retrospective cohort study of 1200 patients and manually collected individual health record data from previous cancer examinations, with retrospective CPA status assessment using international criteria. Among 992 included patients, 16 (1.6%) fulfilled diagnostic criteria for CPA retrospectively, of whom 15 were undiscovered at initial lung cancer examination. The prevalence of CPA in this study population was 50 times higher than the reported prevalence of the overall European population. Our findings indicate that CPA is often missed in patients suspected of malignancy in the chest. Therefore, CPA should be kept in mind as a significant differential diagnosis.

## 1. Introduction

Chronic pulmonary aspergillosis (CPA) is a life-threatening and neglected pulmonary fungal infection. It is estimated that more than three million people are suffering from CPA worldwide [1]; however, evidence on CPA epidemiology is scarce. CPA is still regarded as a rare condition, and the awareness and the knowledge of risk factors for CPA development among clinicians are often limited [2]. Nevertheless, recent studies indicate that the prevalence may be relatively high in selected populations with an estimated 5 year mortality of CPA up to 85% depending on underlying comorbidities and CPA subtype [3,4,5]. Increased awareness of patients at risk, use of paraclinical indicators for early diagnosis, and timely initiation of antifungal treatment are, therefore, crucial to improve the individual patient’s long-term survival. During the COVID-19 pandemic, these factors were only exacerbated as measures to control the pandemic drive drug resistance and more people becoming at risk of CPA [6].

Early diagnosis of CPA is clinically challenging since the variety of symptoms (e.g., breathlessness, sputum production, hemoptysis, malaise, weight loss, low-grade fever) and imaging (e.g., opacity, cavitation) are similar to other common lung diseases (e.g., lung cancer, tuberculosis, chronic obstructive pulmonary disease, emphysema) [1,3,7]. Additionally, these chronic lung disease categories are also independent risk factors for CPA development due to structural lung damage in patients being regarded to possess some degree of immune incompetence. These patients, therefore, might suffer from several severe lung diseases simultaneously [1,5,7].

Due to the overlap in symptoms and clinical presentations, patients with the aforementioned signs and radiological findings may likely be referred for assessment of possible intrathoracic malignancy rather than suspected CPA. At present, no studies have assessed the prevalence of CPA or the optimal CPA diagnostic approach in such a patient population.

In this study, we aimed to (1) assess the prevalence of CPA in a population of patients suspected of chest malignancy, (2) describe characteristics of the used *Aspergillus*-related diagnostic tools, and (3) assess to what extent a CPA diagnosis was missed during the standard assessment of suspected malignancy in the chest when compared to retrospective audit of the electronic patient charts.

## 2. Materials and Methods

### 2.1. Study Design and Setting

We performed a retrospective cohort study of patients with suspected intrathoracic malignancy referred to the Center of Thoracic Oncology (CTO) in the Department of Respiratory Medicine, Odense University Hospital, Denmark, during a 3 year observation period from 2017 to 2019. CTO serves as the primary referral center for the island of Funen (0.5 million inhabitants) and as a tertiary referral center for the Region of Southern Denmark (1.22 million inhabitants). CTO is run by pulmonologists subspecialized in invasive pulmonology and diagnostics of intrathoracic malignancy. Patients for whom CPA is the referral diagnosis are typically assessed in a separate outpatient clinic in the Pulmonary Aspergillosis Center Denmark (PACD) at the hospital.

National Danish guidelines suggest smokers older than 40 years with persistent respiratory symptoms, unexplainable decline in lung function, or unspecific symptoms of malignancy such as malaise, hoarseness, unintended weight loss, or excessive nightly sweating to be examined as part of a so-called “lung cancer package”. Additionally, any patient with imaging findings indicating possible lung cancer may also be referred for the lung cancer package. In the Danish setting, general practitioners and radiologists are, therefore, the primary referrers of patients for lung cancer package diagnostics.

At CTO, depending on referral information, the patient is scheduled for initial computed tomography (CT) or positron emission tomography and CT (PET-CT) as part of the lung cancer package. Further routine examinations consist of patient interview and objective assessment, blood tests, pulmonary function tests (PFT), and various invasive procedures (bronchoscopy, endobronchial ultrasound (EBUS), radial endobronchial ultrasound (rEBUS), esophageal ultrasound (EUS), ultrasound- or CT-guided transthoracic biopsy, thoracocentesis, or diagnostic video-assisted thoracic surgery (VATS)). Following completion of the package, the results are discussed by a multidisciplinary team (MDD) involving pulmonologists, radiologists, specialists in nuclear medicine, oncologists, cardiothoracic surgeons, and pathologists. Cancer treatment is initiated after the MDD.

In addition to the previously mentioned examinations of the lung cancer package, supplementary tests are performed routinely to diagnose relevant differential diagnosis to malignancy. In the study period, blood tests were routinely sent for specific measurement of *Aspergillus fumigatus* IgG and IgE (ImmunoCap, Phadia, Thermo Fischer Scientific, Sweden). Invasive bronchoscopic procedures were supplemented with bronchial lavage (BL) or bronchoalveolar lavage (BAL) for dedicated fungal culture, *Aspergillus* DNA polymerase chain reaction (PCR) testing, and detection of *Aspergillus* galactomannan (GM) (Platelia, Bio-Rad, Hercules, CA, USA). Obtained tissue samples were sent for dedicated fungal culture and *Aspergillus* DNA PCR testing if regarded relevant by the physician performing the invasive procedure. Cultures and PCR testing were performed at the national fungal reference center (Statens Serum Institut (SSI), Copenhagen, Denmark).

### 2.2. Patient Population, Inclusion Eligibility, and Exclusion

In Denmark, the International Classification of Disease edition 10 (ICD-10) is used to register all hospital activity, which includes patients referred for assessment in lung cancer packages [8]. At CTO, the ICD-10 code DZ031B referring to “observation for possible malignancy in the lung” was used for patients assessed in the lung cancer package.

As such patients, were assessed eligible for study inclusion if (A) they had been referred for CTO and (B) the ICD-10 code DZ031B had been registered as part of the examination.

Patients were excluded if (A) they were examined for malignancy at CTO before 1 January 2017, (B) CTO examination for malignancy was ongoing by 31 December 2019, (C) examination was initiated at CTO on benign suspicion, only to later change to malignant suspicion with code DZ031B (“later” defined as examined beyond initial procedures of CT or PET-CT imaging, PFT, patient interview, and blood sampling), (D) they were registered as part of advisory assessment activity between regional hospitals, or (E) other circumstances, such as the patient was referred in error, the patient withdrew consent, or the indication for examination had regressed up to first examination.

### 2.3. Data Sources and Registration

Patient data were collected from the Hospitals Electronic Patient Journal (EPJ) and laboratory results were obtained from the regional hospital departments of radiology, pathology, biochemistry, and microbiology.

The following data were collected: age, sex, initial assessments on referral, exposure to asbestos/work-related lung hazards/tuberculosis/mold, use of narcotics/alcohol/tobacco, PFT, medication, comorbidities, symptoms, biochemistry, CT/PET-CT imaging findings, cytology, histology, microbiology, CTO, and MDD diagnosis when performed. Diverse diagnoses were identified by the use of registered ICD-10 codes in the EPJ, including codes for tentative or verified pulmonary aspergilloses with DB44 (“aspergillosis”). Included patients were followed for relapse/remission of malignancy, as well as additional examinations for cancer, tuberculosis, and CPA until 31 December 2019, distinguishing between examinations continued from CTO outcome or initiated independently. Pre-audit data were limited to patients with first-time examination for malignancy at CTO between 2017 to 2019. Patients who died during or after examination had their date of death registered regardless of whether it was later than 31 December 2019.

For data collection and storage, we used REDCap hosted by the Odense Patient Data Exploratory Network (OPEN), using a survey format of close-ended questions [9,10]. Where applicable, REDCap data points were subject to real-time validation, using forced data formats and user alerts for unexpected entries (i.e., numerical outliers). To anticipate missing data caused by human error, the survey would alert the user if an empty field was saved. Relevant data not fitting into the fields were stored as free text and reviewed ad hoc to expand survey options in existing questions. Inbuilt data quality tools were used ad hoc to monitor and correct issues, including unfilled entries.

### 2.4. Pre-Audit Screening

The primary investigator (R.R.), and five assistants manually collected patient information from the EPJ. In cases with multiple sets of data, the following attributes were prioritized: (1) results produced by CTO, (2) minimal deviation from CTO visitation date, and (3) completeness of dataset. The study aimed to maximize the study size within the planned data collection period.

The study phases are illustrated in Figure 1. If one or more criteria for audit were fulfilled in the pre-audit data collection, the patient would proceed to audit. The criteria for audit were one or more of the following:Pulmonary aspergillosis of any kind diagnosed by CTO;Pre-audit reviewer request audit due to doubt of audit eligibility;Imaging (CT/PET-CT):∘imaging report suggesting *Aspergillus*-related disease;∘identified cavity, necrosis, or pulmonary aspergilloma;
Pathology report:∘suggest fungal pathogenesis;∘describe detection of fungal elements;∘identify cavities, necrosis or pulmonary aspergilloma;Microbiology:∘fungal elements detected by microscopy;∘positive culture of any *Aspergillus* species;∘positive PCR of *Aspergillus* DNA;∘BL/BAL *Aspergillus* GM titer >0.6;Serology: (blood)∘IgE > 1000 × 10^3^ IU/L;∘eosinophilic granulocytes > 0.5 × 10^9^/L;∘*Aspergillus fumigatus* IgG > 75 mg/L;∘*Aspergillus fumigatus* IgE > 0.35 × 10^3^ IU/L;∘*Aspergillus niger* IgG > 50 mg/L;∘*Aspergillus* IgG ≥ 1.5 AU/mL;∘*Aspergillus* GM titer > 0.6;∘other specific *Aspergillus* test performed.

### 2.5. Final CTO Diagnosis

As part of the pre-audit screening, the final diagnosis after completion of the lung cancer package was also recorded. The MDD diagnosis or, if no MDD was performed, the diagnosis recorded by a pulmonologist from the CTO upon completion of the lung cancer package was used as the final diagnosis. If CPA or another *Aspergillus*-related disease was registered as the final diagnosis, the relevant subtype recorded in the EPJ was also noted.

### 2.6. Audit and Reference CPA Diagnostic Criteria

The audit process consisted of separate assessments by two specialists (J.R.D., C.B.L.) aimed at obtaining consensus on CPA using predefined diagnostic criteria. In case of disagreement, a predesignated third specialist (F.R.) would make the ruling consensus decision. The auditing experts had access to EPJ information and were not limited by the scope of the pre-audit data collection.

The following predefined diagnostic criteria were used for CPA [1]:One or more cavities with or without a fungal ball present or nodules on thoracic imaging;Any direct or indirect mycological evidence from respiratory samples or from blood of *Aspergillus* spp. Infection;Exclusion of an alternative diagnosis;Disease present for at least 3 months.

If the patient fulfilled all four criteria, the patient was diagnosed with CPA.

### 2.7. CPA Subtype

For patients in which the CPA diagnostic criteria were met, the auditors subsequently made a consensus determination of the CPA subtype using the principles and criteria described by Denning et al. [1]. The patients were, thus, divided into the following subtypes:*Aspergillus* nodule(s);Simple aspergilloma;Chronic cavitary pulmonary aspergillosis (CCPA);Chronic fibrosing pulmonary aspergillosis (CFPA);Subacute invasive aspergillosis (SAIA).

In the case of possible overlap between subtypes or subtype conversion during follow-up, the auditors chose the subtype being the clinically dominant at the time of referral to CTO.

### 2.8. Non-CPA Aspergillus-Related Lung Disease

If the patient did not fulfill CPA criteria, the auditors additionally assessed whether the patient met criteria for other forms of *Aspergillus*-related pulmonary disease (e.g., allergic bronchopulmonary aspergillosis (ABPA), severe asthma with fungal sensibilization (SAFS), invasive pulmonary aspergillosis (IPA), or *Aspergillus*-related hypersensitivity pneumonitis (HP)) in accordance with diagnostic criteria as defined in international recommendations) [2,11,12,13].

### 2.9. Statistical Analysis

The main outcome was the estimated 3 year prevalence of CPA, defined by the quantity of audit-verified CPA cases (and, thus, independent of possible CTO aspergillosis diagnosis) divided by the quantity of pre-audit study participants.

Secondary outcomes were the prevalence of specific patient characteristics associated with CPA, prevalence of other *Aspergillus*-related diseases, and prevalence of diagnostic tools related to CPA.

Continuous numerical variables were expressed as medians with interquartile ranges (IQR), while categorical variables were expressed as absolute frequency *N* and percentages (%).

The pre-audit population and the subpopulations of non-CPA and confirmed CPA were analyzed separately.

All analyses were performed using STATA16 (StataCorp LLC, College Station, TX, USA).

## 3. Results

A total of 1200 patients were eligible for inclusion, of which 208 were excluded on the basis of initial screening. Manual data collection was performed for 992 patients as part of the pre-audit screening with an additional 14 exclusions after full data collection. The pre-audit screening identified 220 patients meeting one or more of the predefined audit criteria. Out of 220 patients, 16 (1.6%) met the criteria for CPA, while four patients were diagnosed with ABPA and one patient was diagnosed with IPA. The study flowchart is presented in Figure 1 with a breakdown of exclusion reasons, detailed in Section 2.

Of the audited patients, 26 fulfilled only three out of four criteria for CPA, in whom four lacked a characteristic radiological pattern, five lacked mycological evidence of *Aspergillus*-species, 11 lacked exclusion of alternative diagnosis, and six did not have persisting disease for more than 3 months.

At baseline, 54% were males with a median age of 70 (IQR 62–76). Of all study participants, 78% had a history of tobacco use, and more than half had a history of cardiovascular disease. The dominant symptoms were coughing, dyspnea, and loss of weight. Stratification to pre-audit, non-CPA and CPA groups yielded no clear distinction at baseline. (Table 1, Table A1).

CT imaging and fluordeoxyglucose (FDG)-PET/CT were available for almost all participants as part of CTO examination. Almost all nondiffuse opacities were FDG-positive independent of size, including those concerning the confirmed CPA group. None of the imaging findings were specific for CPA, and the majority of patients with nodules or opacities with cavitation did not have CPA (Table 2). No patients were described with either “aspergilloma” or “fungal ball”.

Considering laboratory tests, the CPA group had higher *Aspergillus fumigatus* IgG levels than the non-CPA group. Despite this, the majority of patients with an *Aspergillus fumigatus* IgG level of above 75 mg/L still did not meet the criteria for CPA. None of the other laboratory results were specific for CPA, with the majority of positive results being from the non-CPA group. No other tests were able to distinguish between CPA and non-CPA patients. (Table 3). In the CPA population, one patient had a positive interferon-gamma release assay, but acid-fast staining, mycobacterial PCR, and cultures remained negative on respiratory samples. In addition, no patients with CPA tested positive for human immunodeficiency virus.

A total of 193 (19%) patients were dead within 1 year of CTO inclusion; however, among the confirmed CPA patients, none died within 1 year. A total of 272 (27%) individuals were diagnosed with primary lung cancer by CTO, and 48 (5%) individuals were diagnosed with metastatic cancer disease spreading to the lungs. Four individuals were diagnosed with any type of *Aspergillus* disease by CTO, and 157 (16%) were diagnosed with other diseases such as sarcoidosis and pneumonia.

During the prospective diagnostic assessment in CTO, one patient was diagnosed with ABPA, and three were diagnosed with pulmonary aspergillosis without specification. Of these patients with unspecified aspergillosis, one was audited as having ABPA, one was audited as having CCPA, and one was audited as having “possible *Aspergillus* bronchitis”. Thus, 15 of 16 CPA cases in the audit were initially undiagnosed during the CTO assessment.

## 4. Discussion

### 4.1. Key Results

Of 978 patients examined for lung cancer, 16 patients were retrospectively diagnosed with CPA during a systematic audit using predefined criteria, corresponding to an estimated 3 year CPA prevalence of 1.6%. Of all collected patient characteristics and individual tests, only *Aspergillus fumigatus* IgG suggests an ability to identify CPA patients, albeit insufficient on its own. Only one of the 16 CPA patients was diagnosed at the time of CTO.

### 4.2. Interpretation

The findings correspond to a CPA ratio of one in 61 examined, and, while prevalence data of CPA in the overall Danish population are unavailable, the estimated European all-population CPA ratio is 1 in 3100 (i.e., a factor of 50 difference) [1,14]. In addition to CPA, five patients were found to have other types of pulmonary aspergillosis: ABPA (*n* = 4) and IPA (*n* = 1), leading to a total *Aspergillus*-related disease prevalence of 2.1%. Notably, no patients with CPA died within 1 year of examination at CTO, suggesting a better prognosis in this population than previously described [5].

To our knowledge, this is the first study to examine CPA prevalence among patients referred for clinically or radiologically suspected pulmonary malignancy. As the diagnostic criteria used for CPA during this audit dictate exclusion of other disease, an eventual concomitant CPA diagnosis was withdrawn in patients diagnosed with other verified diagnoses due to CTO examination (e.g., lung cancer, sarcoidosis). This may result in an overall underestimation of the CPA prevalence, as well as delayed treatment among patients with concomitant disease, and a risk of severe *Aspergillus*-related complications in immunocompromised patients. A total of 11 cases fulfilled three of four criteria for CPA but lacked exclusion of alternative disease. However, whether these consist of true concomitant disease is unknown as the radiological signs were not specific for one disease; furthermore, *Aspergillus* species are commonly found as contaminants from the environment and upper airways.

Comparing CPA patients with non-CPA patients, this study can only demonstrate *Aspergillus fumigatus* IgG to be able to distinguish between groups, albeit not specific for CPA at an individual level. As *Aspergillus fumigatus* IgG contributes substantially to the fulfilment of the mycological criterion for CPA, it is expected that many of the identified CPA patients surpassed the threshold of >75 mg/L; however, 65 (12%) of the non-CPA group did as well and, therefore, the test is unable to stand alone. The chosen *Aspergillus fumigatus* IgG threshold was based on local laboratory recommendations, but an approach using some of the lower cutoff values previously reported in the literature could also have been used. This would possibly have led to an increased number of patients meeting CPA diagnostic criteria, but would also most likely have led to a substantial increase in patients registered as having elevated *Aspergillus fumigatus* IgG levels but not CPA. Since neither screening nor diagnostic criteria were solely based on elevated *Aspergillus fumigatus* IgG levels, we do, however, believe the potential number of CPA patients being misclassified as not having CPA to be very low. Generally, our results carefully support the consensus that clinical, radiological, and laboratory signs of possible CPA are likely to be unspecific, and diagnosis is reliant on appropriate use of a combination of these unspecific signs and tools. Further prospective diagnostic accuracy studies are needed to address the optimal value and combination of the CPA diagnostic tools used in this population, particularly which tests may be used as initial CPA screening tools in patients being assessed for possible malignancy.

The notion that awareness of CPA is low, even in a highly focused cancer-oriented setting, is supported by our finding that 15 of the 16 CPA cases in the audit were undiagnosed during CTO assessment. Without consistent and appropriate diagnosis of other lung infections rather than only determining cancer status, CTO may have misdiagnosed some cases of CPA as pneumonia [2], prompting further diagnostic and potential treatment delay. This study also found that specific *Aspergillus fumigatus* IgG was only measured for half of the examined patients despite this tool being part of the routine regime for all patients examined as part of the lung cancer package, possibly due to the many ways the patients could enter the lung cancer package with varying degrees of already performed tests and serology analysis, of which some were only available at CTO. Similarly, any culture or PCR analysis of *Aspergillus* species in biopsy, sputum, and lavage procedures was performed for one-quarter of the population, citing reasons such as the clinician not finding indication for the laboratory analysis or sampling not being viable in the situation. As such, some CPA patients may have been missed entirely in the clinic, as well as underreported in this study, due to insufficient coverage of the examination regime.

### 4.3. Strengths and Weaknesses

As this study was conducted at a single center, the external validity for the Danish and European population may be limited. CTO is the largest center for lung cancer in the Region of Southern Denmark, but other centers in the region may disproportionately service rural communities, and a higher proportion of “complicated” cases may be referred and accumulated at CTO, rather than be distributed by geographic origin. Conversely, the inclusion criteria did not contain demographic restrictions, thereby ensuring the study population reflected the actual population referred to CTO, as well as permitting high external validity among patients at risk of lung cancer nationwide, as entry to the lung cancer package across regions is based on the same national guidance.

A significant consideration regarding the radiological findings is that PET/CT was put ahead of CT-only imaging in cases where both types of examination had been performed. It is likely that PET-CT examinations performed in continuation of CT underreport specific CT findings such as “aspergilloma” or “fungal ball” or simply report such findings as generic “opacity”, skewing our results to underreport these types of radiological findings.

It is our assessment that the risk of misclassification stemming from the criteria for audit eligibility is low as criteria were numerous and broad, any single criterion was sufficient to qualify for audit, and the data collectors were allowed to force audit if in doubt. This provided a consistent screening and inclusion, thereby reducing intra- and interobserver variance. Of the pre-audit population, 78% were not eligible for audit, and, of the audited records, 90% were not diagnosed as having any aspergillosis, resulting in a homogeneous and steep “narrowing funnel” across study phases, giving us confidence that the combined criteria were not overly restrictive, as, for each case of CPA or other form of aspergillosis, many potential cases were evaluated.

## 5. Conclusions

The prevalence of CPA among patients suspected of lung cancer was 1.6%. As our audit could identify several missed CPA cases, we propose increased organizational efforts to improve the awareness of possible CPA among clinicians. In addition, as none of the used diagnostic tools were specific for CPA in this patient population, further studies assessing optimal diagnostic screening algorithms are warranted.

## Figures and Tables

**Figure 1 jof-08-00297-f001:**
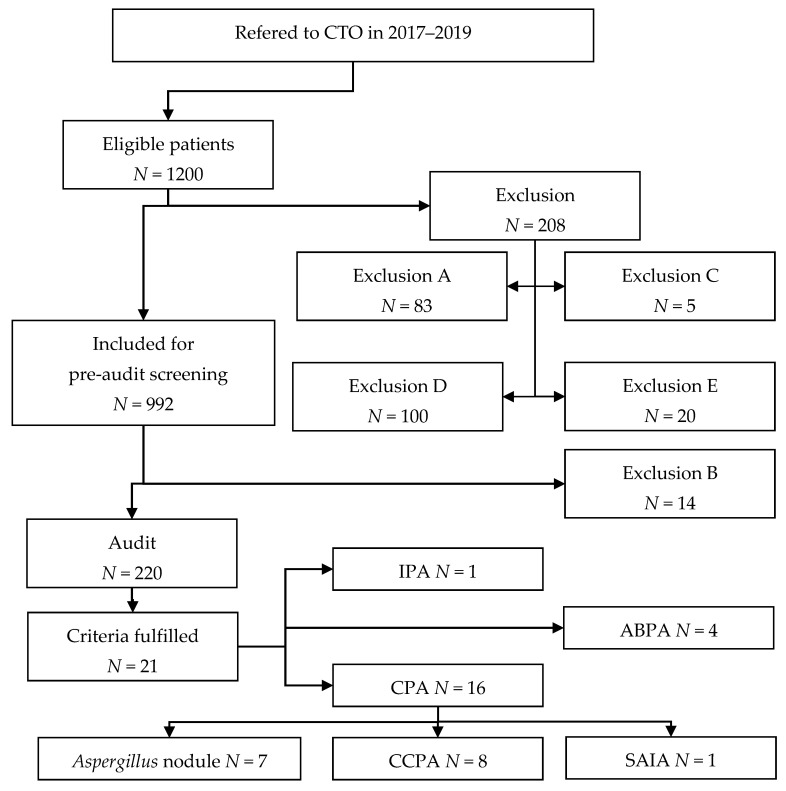
Study flowchart. Exclusion types are detailed in Section 2.2. See Donnelly et al. for definition of probable IPA [11].

**Table 1 jof-08-00297-t001:** Baseline characteristics.

Variable	All Included Patients	Non-CPA	CPA
Demographic data	* N * = 978	* N * = 962	* N * = 16
Male	532	524	8
Female	446	438	8
Age (IQR), years	70 (62–76)	70 (62–76)	66 (58–77)
BMI (IQR), kg/m^2^	24.8 (21.6–28.4)	24.7 (21.5–28.4)	25.9 (22.2–28.5)
Exposure			
Alcohol *N* (%) units per week ≥7 (♀)/≥14 (♂)	207/799 (26)	204/786 (26)	3/13 (23)
Smoker, ever, *N* (%)	760/936 (81)	745/920 (81)	15/16 (94)
Tobacco package years Median (IQR)	35 (20–50)	35 (20–50)	39 (25–50)
Symptoms *N* (%)	* N * = 978	* N * = 962	* N * = 16
Dry cough	489 (50)	479 (50)	10 (63)
Cough with sputum	248 (25)	243 (25)	5 (31)
Hemoptysis	53 (5)	52 (5)	1 (6)
Dyspnea	356 (36)	350 (36)	6 (38)
Fatigue	289 (30)	284 (30)	5 (31)
Fever	56 (6)	55 (6)	1 (6)
Weight loss	336 (34)	332 (35)	4 (25)
Loss of appetite	99 (10)	97 (10)	2 (13)
Night sweats	119 (12)	115 (12)	4 (25)
Asymptomatic	74 (8)	74 (8)	0
Comorbidity *N* (%)	* N * = 978	* N * = 962	* N * = 16
None	73 (8)	73 (8)	0
Previous or current mycobacterial disease	11 (1)	11 (1)	0
COPD	225 (23)	222 (23)	3 (19)
Cardiovascular disease	489 (50)	484 (50)	5 (31)
Diabetes mellitus	123 (13)	121 (13)	2 (13)
Previous venous thromboembolism	62 (6)	60 (6)	2 (13)
Previous lung cancer	13 (1)	12 (1)	1 (6)
Previous or current verified non-lung malignancy	202 (21)	197 (20)	5 (31)
Immunosuppressive disease	5 (1)	5 (1)	0
Previous thoracic surgery	22 (2)	22 (2)	0
Medication ^A^ *N* (%)	* N * = 978	* N * = 962	* N * = 16
No medication	58 (6)	58 (6)	0
Inhaled steroids	141 (14)	138 (14)	3 (19)
Oral steroids	73 (7)	71 (7)	2 (13)
Other immunosuppressive drugs ^B^	65 (7)	61 (6)	4 (25)
Antibiotics	112 (11)	110 (11)	2 (13)
Systemic antifungal drugs	20 (2)	20 (2)	0

^A^ Prescription drugs used by the patient, verified as part as CTO patient interview. ^B^ Cytostatics, anti-metabolites, macrolides, antineoplastic drugs, immune-modulating drugs, anti-CD20, anti-interleukin 6 (anti-IL6), anti-tumor necrosis factor, protein kinase inhibitors, and chemotherapy treatment.

**Table 2 jof-08-00297-t002:** Imaging.

Variable	Pre-Audit	Non-CPA	CPA
Lung parenchyma *N* (%)	* N * = 974	* N * = 958	* N * = 16
Nodule (<30 mm)	478 (50)	469 (49)	9 (56)
- Reticular	86 (9)	83 (9)	3 (19)
- Cavitation	21 (2)	20 (2)	1 (6)
- FDG-positive	476 (49)	467 (49)	9 (56)
Mass/tumor (≥30 mm)	217 (22)	215 (22)	2 (13)
- Reticular	21 (1–3)	20 (1–3)	1 (6)
- Cavitation	19 (1–3)	17 (1–3)	2 (13)
- FDG-positive	217 (22)	215 (2)	2 (13)
Diffuse opacity of uncertain size	149 (15)	143 (15)	6 (38)
Pleural effusion	120 (12)	118 (12)	2 (13)
Lymph adenopathy *N* (%)	* N * = ^A^	* N * = ^B^	* N * = ^C^
Mediastinal nodes			
- CT-visible adenopathy	210 (21)	208 (22)	2 (13)
- Lymph FDG-positive	320 (35)	313 (34)	7 (47)
Hilar nodes			
- CT-visible adenopathy	147 (15)	146 (15)	1 (6)
- Lymph FDG-positive	270 (29)	264 (29)	6 (40)
Peripheral lung nodes			
- CT-visible adenopathy	26 (3)	26 (3)	0
- Lymph FDG-positive	29 (3)	29 (3)	0
Extrapulmonary nodes ^D^			
- CT-visible adenopathy	44 (5)	44 (5)	0
- Lymph FDG-positive	93 (10)	92 (10)	1 (7)

^A^ CT-imaging data available for 974 and FDG-PET/CT imaging available for 925 patients. ^B^ CT-imaging data available for 958 and FDG-PET/CT imaging available for 910 patients. ^C^ CT-imaging data available for 16 and FDG-PET/CT imaging available for 15 patients. ^D^ Axillar, low cervical, supraclavicular, sternal notch, parasternal, and superior diaphragm.

**Table 3 jof-08-00297-t003:** Pulmonary function and laboratory tests.

Variable	Pre-Audit	Non-CPA	CPA
** Spirometry ** median (IQR)	** * N * = 869 **	** * N * = 853 **	** * N * = 16 **
FEV1, L	2.1 (1.5–2.7)	2.1 (1.5–2.7)	2 (1.6–2.8)
FEV1, % predicted	81 (61–98)	81 (61–98)	78 (63–96)
FVC, L	3.2 (2.5–3.9)	3.2 (2.5–3.9)	3 (2.4–4)
FVC, % predicted	96 (79–112)	96 (79–112)	94 (85–111)
FEV1/FVC index	69 (59–76)	69 (60–76)	67 (54–77)
** Serology tests **			
** * Aspergillus fumigatus * IgG **	** * N * = 577 **	** * N * = 563 **	** * N * = 14 **
Median (IQR), mg/L	25.9 (12.5–50.9)	25.1 (12.2–47)	115 (80.7–160)
>75 mg/L (%)	76 (13)	65 (12)	11 (79)
** * Aspergillus fumigatus * IgE **	** * N * = 586 **	** * N * = 572 **	** * N * = 14 **
Median (IQR), 10^3^ IU/L	<0.1 (-)	<0.1 (-)	<0.1 (-)
>0.35 × 10^3^ IU/L, *N* (%)	30 (5)	30 (5)	0
** Microbiology *N* (%) **			
*Aspergillus* culture	12/255 (5)	8/245 (3)	4/10 (40)
- Sputum	4/54 (7)	3/52 (6)	1/2 (50)
- BL/BAL	7/180 (4)	5/172 (3)	2/8 (25)
- Lung tissue biopsy	1/71 (1)	0/68 (0)	1/3 (33)
*Aspergillus* PCR	6/132 (5)	5/124 (4)	1/8 (13)
- BL/BAL	5/127 (5)	4/119 (3)	1/8 (13)
- Biopsy	3/23 (3)	3/23 (13)	0/0 (0)
BL/BAL *Aspergillus* GM titer >0.6	10/103 (10)	8/94 (9)	2/9 (22)
** Histopathology *N* (%) **			
Fungal hyphae/debris in tissue	4/35 (11)	2/32 (6)	2/3 (67)

## Data Availability

The data presented in this study are available on request from the corresponding author. The data are not publicly available due to Danish legislature. Data access permission can be obtained for specific purposes.

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
