# Peer review of "Prevalence of Chronic Pulmonary Aspergillosis in Patients Suspected of Chest Malignancy"

_jof, 2022, doi:10.3390/jof8030297_

Round 1

Reviewer 1 Report

It is an interesting topic and give som usefull information for the clinician, despite it corroborates what is already known.  The method is clearly described and easy to read.  The discussion underscore perfectly the limitation of the study. 

There is no information regarding the fact that the study was submitted to ethical comittee. And what about the retrospective use of patients data.  

Author Response

In accordance with current Danish legislation, the study was not required to obtain separate ethics committee approval since it was a retrospective study. These studies are typically approved by the “Danish Patient Safety Authority”. The section has been slightly altered to clarify these aspects.

Reviewer 2 Report

Authors wrote an interesting and well wrote paper. I find it relevant for scientific community. It is not easy for me give some suggestions because the paper is very well presented and the research question is very relevant.

Below my minor suggestions:

  1. Introduction: add also that during COVID 19 pandemic fungi infection, aspergillum infection increase their already relevant role and resistance. In fact, critically ill COVID-19 presents several major risk factors for invasive fungal infections, such as mechanical ventilation, prolonged ICU stays, older age, protracted corticosteroid therapy, and extensive antimicrobial exposure(see and cite Impact of SARS-CoV-2 Epidemic on Antimicrobial Resistance: A Literature Review. Viruses. 2021 Oct 20;13(11):2110)
  2. Methods and results: clear and well presented
  3. Discussion: have you found any other concomitant infections?
  4. Conclusion: give some proposal that came from your interesting data

Author Response

Thank you for the very positive and constructive reviewer comments.

1: Since the project was conceptualized before the COVID-19 pandemic, no patients or outcomes were affected by the COVID-19 pandemic. The study setting is an outpatient setting assessing prevalence of chronic pulmonary aspergillosis in patients with suspected malignancy in the chest. The setting and population are thus significantly different than a population of critically ill COVID-19 patients. But since awareness of pulmonary aspergillosis is indeed highly important in this patient population, a sentence mentioning the aspects of the COVID-19 pandemic has been inserted with a reference to the study suggested by the reviewer.

3: Since the diagnostic criteria used to define CPA in the study involved exclusion of relevant differential diagnosis, all cases diagnosed with CPA or other Aspergillus related diagnosis were not diagnosed with concomitant infections. The collected data does not permit a reliable count of other infections, a similar study would need to be performed to get reliable data on for example mycobacteria infections. In our data analysis we did however not encounter any disease “trends” found to be noteworthy including concomitant infections. Since our study is unable to address the clinically relevant question raised by the reviewer and we did not recover any patients in our data analysis we chose not to address these aspects in the conclusion and has as such not altered this approach in the discussion.

4: Conclusion: As mentioned in the conclusion our data and results does not allow us to propose a specific optimal diagnostic approach. The conclusion has been slightly modified to address that organizational efforts ensuring increased clinical awareness are still warranted.

Reviewer 3 Report

good job!, really nice article.

Author Response

Thank you for the very positive comments.